# The Socioeconomic Dimensions of Water Scarcity in Urban and Rural Mexico: A Comprehensive Assessment of Sustainable Development

**Silvana Pacheco-Treviño *** and **Mario G. Manzano-Camarillo**

School of Engineering and Sciences, Tecnológico de Monterrey, E. Garza Sada 2501 Sur, Monterrey 64849, Mexico; mario.manzano@tec.mx
* Correspondence: silvanapacheco14@gmail.com

**Abstract:** Mexico faces severe water scarcity due to population growth, industrial activities, and climate change. The arid and semidesert conditions prevalent in northern Mexico, particularly in Nuevo Leon, significantly accentuate the challenges associated with water scarcity. This region is vulnerable to water scarcity due to minimal rainfall, recurrent droughts, and the increasing pressure of water demand from the densely populated Monterrey. We examined the disparities that contribute to water poverty by comparing water scarcity between rural and urban populations in Nuevo Leon. The results revealed significant contrasts in water scarcity between the two populations, indicating that different factors contribute to water poverty based on regional, territorial, and cultural characteristics. We selected the water poverty index (WPI) as an evaluation metric due to its inherent compatibility with available data sources, which facilitates its application to stakeholders and ensures comparability with other regions. This study contributes to studies on water scarcity assessment by addressing a critical limitation of the WPI. We compared three weighting methods—equal weight, principal component analysis (PCA), and analytic hierarchy process (AHP)—and identified that PCA and AHP demonstrated a superior performance compared to the standard methodology. These findings underscore the importance of considering region-specific conditions, as well as socioeconomic disparities between rural and urban populations and their role in vulnerability to water scarcity in calculating water poverty. These insights provide valuable information for customized solutions to regional challenges, representing leading actions toward sustainable development.

**Keywords:** water poverty; Mexico; Nuevo Leon; scarcity; sustainable development; SDG 6

## 1. Introduction

Water scarcity is emerging as a persistent and formidable issue in Mexico, which affects both urban and rural areas. Water availability in the country faces constant threats emerging from resource competition, pollution, and the effects of climate change [1–3]. Population growth exacerbates the demand for water resources [4]. Moreover, climatic and geographical conditions escalate the process of desertification in arid and semiarid regions. Against this background, scholars project that the consequences of climate change will include a significant reduction in water availability due to increasingly frequent and severe droughts [5].

Presently, Mexico is facing an extreme drought situation, which leads to acute water shortages in nearly two thirds of municipalities [6,7]. The gravity of this situation pushed several cities in 2022 to the brink of a day-zero scenario, wherein the available water supply is unable to meet the demands of the population [8,9]. The main cities in the country are strongly dependent on the water resources of local water basins for their socioeconomic development, leading to a decrease in water availability due to overexploitation. Monterrey, the second-largest city in the country, has faced severe problems with water supply due to water stress, scarcity, and quality issues [6,10,11].

Estimations indicate that, by 2030, the water availability in northern Mexico will decrease to less than 1600 m$^3$ per person per year, which represents a substantial decline from the 3000 m$^3$ per year recorded in 1960 [12]. This decline is primarily attributed to fluctuations in precipitation patterns, however, water scarcity is not exclusively tied to the physical constraints of water availability [13]. This reduction in water availability can be attributed to social factors, including land use, water management practices, regulatory frameworks, demographic shifts, and pollution, which, along with climatic constraints, further contribute to water scarcity [14,15]. Therefore, such factors are directly linked to actions to which stakeholders and policy planners can direct their attention [16,17]. The implementation of sustainable water governance and management practices is deemed to play a substantial role in addressing the water scarcity crisis [14,15,18–20].

This current scenario of impending scarcity will unavoidably restrict water access and pose a significant threat to the attainment of the Sustainable Development Goals (SDGs) established by the UN. If the challenge to achieve these goals is ambitious today, it may become insurmountable in the next decade [21,22]. Given the last Report on Climate Change, disturbing outcomes are anticipated if profound actions toward integral water management are not implemented in the current decade [5].

The arid and semidesert conditions in northern Mexico, particularly in the state of Nuevo Leon, significantly exacerbate the challenges associated with water scarcity [23]. This region is fundamentally vulnerable to water scarcity due to minimal rainfall, recurrent drought, and limited water resources. The fact that a large proportion of the population resides in Monterrey, the capital of Nuevo Leon, compounds these natural challenges. This high population density has intensified the severity of water shortages, especially in the last three years [9,11].

With a population exceeding five million, the Monterrey Metropolitan Area (MMA) relies heavily on groundwater aquifers and surface reservoirs to meet its water needs [8]. Moreover, rapid urban growth over the last decade has accelerated the depletion of these aquifers, which has resulted in the drying of soil moisture and the persistence of drought conditions. Moreover, the MMA has a strong manufacturing sector and has been frequently referred to as the industrial capital of Mexico. These industrial activities place additional pressure on the previously strained water resources in the region [24,25].

The escalating demand for water resulting from these demographic trends has surpassed the capacity of the existing water systems to meet the needs of the urban population. The consequences of drought impose restrictions on the use of the remaining water resources. However, the rural population is facing additional challenges due to limited access to clean water sources [26]. As such, obtaining water in remote and semidesert areas is frequently accompanied by high economic costs [17].

The absence of adequate water services faced by the rural population has exerted a detrimental effect on their overall progress, which hinders social development. Water scarcity creates a domino effect that impacts various aspects of rural life, such as health, education, livelihoods, and general resilience [27–30] Specifically, rural communities in southern Nuevo Leon encounter significant shortcomings in terms of access to drinking water and sanitation services [7,15,31,32]. This difficult situation perpetuates the cycle in which rural populations continue to experience poverty and marginalization. Thus, addressing the acute water crisis faced by rural communities is imperative to ensure that it is not overshadowed or neglected by the pressing urgency of addressing the water crisis in the city [33,34].

The most vulnerable population to water scarcity in Mexico live in the rural communities of arid regions scattered in isolated locations [26]. This geographic dispersion poses challenges for public institutions in terms of providing quality public services, which results in increased costs for government investments. Compounding this issue, climate change exacerbates the frequency and severity of drought in arid regions [31].

Information regarding the extent of pressure on water resources faced by the rural population of Nuevo Leon is limited [23–25]. In these communities, water allocation

for human consumption, agriculture, and domestic activities is predominantly reliant on overexploited aquifers and a limited number of locally constructed pond reservoirs that capture scarce rainwater. The agricultural use of water further diminishes the available water in quantity and quality for human consumption in rural areas [20,26].

Overexploited aquifers and stored water sources are susceptible to rapid deterioration in quality, which poses substantial health risks to the local population [10,27,30]. Hence, the persistent shortage of water for human consumption poses a persistent issue that is anticipated to intensify due to the influence of climate change conditions and the increasing demands of urban centers [1,3,5,35]. If drought conditions are exacerbated, these communities may experience elevated socioeconomic setbacks, including financial, temporal, and political trade-offs, in their pursuit of water supply, such as a reliance on water trucks, bottled water, or long-distance water transportation [21,27,36].

Although each region has unique experiences regarding the water crisis according to local environmental, social, and economic conditions, rural and urban populations in Nuevo Leon are predominantly situated in a semiarid region characterized by limited precipitation and prolonged periods of drought. Although no direct correlation is observed between poverty and water availability, the extent of accessible water resources profoundly influences the capacity of communities to meet essential requirements [23,24,29]. Several studies have emphasized the significance of considering the influence of socioeconomic factors on water availability [13,16,27,37]. Recognizing socioeconomic disparities between rural and urban populations and developing tailored solutions is essential for addressing their distinct challenges [38].

Common proposed measures to alleviate water scarcity encompass the allocation of resources toward the enhancement of water infrastructure, distribution, and sewerage systems, as well as the implementation of initiatives that target reforestation and improvements in land management practices [39]. However, the limited availability of financial resources presents a significant challenge to the effective implementation of these proposed solutions, necessitating the prioritization of projects that yield maximum benefits for most of the population. In response to drought severity, the Mexican government declared a national emergency in 2022, which directed increased financial resources toward improvements in water infrastructure and the mitigation of water demand in the urban center of Monterrey [11,40,41].

Regrettably, less attention has been given to addressing the pressing water crisis experienced by rural communities located in the southern region [31]. Striking a complex balance between economic activities, such as industrial agriculture and manufacturing, and ensuring adequate water access are of paramount importance to prevent the exacerbation of the water crisis in the region.

The importance of water resource management becomes increasingly significant when considering the compounding effects of climate change [35,42]. The escalating consequences of rising temperatures, shifting precipitation patterns, and the increasing frequency of extreme weather events impose formidable challenges on the availability of water. Given these circumstances, the depletion of water resources is becoming an imminent concern, which necessitates prompt and decisive measures for constructing water infrastructure and establishing regulations on private usage in agriculture and industries. Failure to undertake these actions expeditiously would impede the achievement of SDG 6: equitable access to safe water [37,43].

This study uses the water poverty index (WPI) to evaluate the situation within the study area, identify key factors that require attention for improvement in water management, and emphasize the urgent measures necessary for attaining SDGs [14,22]. We propose the WPI as a versatile and replicable model for evaluating the causes and impacts of water scarcity and its relation to poverty and socioeconomic conditions. By employing the WPI, studies have evaluated water scarcity and vulnerability at different scales and seasons, national or regional [6,23,31,44–48]. Despite these evaluations demonstrating the

significance of socioeconomic conditions in water scarcity, comparability is hindered by variations in methodology and scale.

Past studies have recognized two major limitations associated with the WPI [49,50]. First, calculations using available data are typically problematic because the majority of estimates are adjusted to suit the required data. This limitation can be avoided by using satellite remote sensing data and rapid advancements in big data methods, which provide different scales and long-term monitoring approaches [51].

Further, a common practice in the majority of WPI calculations is assigning equal weights to its components. Consequently, the current study explores two alternatives concerning the weights of components by employing multivariate techniques. However, a precise evaluation and its impacts on communities persist as a great challenge [13]. A significant gap continues to exist between the ability to acquire water-related data and knowledge related to water and the practical need for monitoring and evaluation, especially in developing countries [50].

Applying the WPI at the municipal level in Mexico, which is the smallest administrative unit, is crucial, because it can provide relevant and actionable insights, which contribute to the identification of the most marginalized communities struggling with challenges of water scarcity at the regional scale. By identifying them, the findings can serve as a significant reference for policymakers and stakeholders and increase awareness of their roles in addressing the challenge of achieving the SDGs by 2030 [39,43]. As a solution at the national level, this methodological proposal can be replicable and comparable.

Moreover, this analysis facilitates an in-depth understanding of the intricate relationships among poverty, social marginalization, ecological integrity, water availability, and health. This understanding is particularly beneficial given the comprehensive nature of the SDG 6 indicator framework, which encompasses multiple targets [43]. Information generated from this research can guide decision-making processes and efforts to develop projects and programs that promote equitable water access and sustainable development.

## 2. Materials and Methods

### 2.1. Study Area

The state of Nuevo Leon, situated in northeastern Mexico, is strategically important for the national economy because of its close commercial ties with the United States and substantial industrial activity. Approximately 88% of the population resides in the capital city of the MMA. With a population exceeding 5 million individuals, the MMA exhibits a high population, which has nearly doubled in 30 years (1990–2020; [52]). The remaining population, that is, the rural communities, resides sparsely and scattered throughout the region [7,12,33,53].

The MMA exhibits distinct socioeconomic and environmental characteristics. It has undergone rapid urbanization and population growth and serves as a prominent industrial center. The manufacturing sector, which contributes to the economic vitality of the area, places a significant strain on water resources [11]. Groundwater aquifers and surface reservoirs are the primary sources of water supply; however, their exploitation has resulted in depletion, which exacerbates drought conditions [41]. Consequently, the region faces challenges in meeting the escalating water demand, which leads to shortages and inadequate access to clean water and sanitation services [8,24,54].

Given the geological and hydrological characteristics of the region, along with projections of climate scenarios that indicate reduced water availability in the future [35], the state of Nuevo Leon is primarily composed of arid ecosystems that occupy approximately 87.3% of the land area [24]. A comparatively small portion (6.7%) is dedicated to forest ecosystems [24]. Considering the expected climate variability and potential exacerbation of extreme events, adaptation strategies are necessary to cope with evolving conditions [35,42]. Moreover, increased investment in the manufacturing and industrial sectors and the continuous population growth of the MMA pose significant pressure on and competition for water resources among the rural population [11,54].

Maintaining the current clean and safe water supply is a major challenge for the city and the state of Nuevo Leon. Although the MMA is notable for its relatively high coverage rates of water and sanitation services, rural communities in the southern region of Nuevo Leon exhibit below-average coverage of essential services, such as drinking water, sanitation, and sewage treatment, compared with the national average. These marginalized zones predominantly compromise small settlements characterized by a low proportion of the municipal population [52]. Furthermore, this region experiences frequent and severe drought periods [31]. These circumstances highlight the vulnerability of the area to water scarcity.

In the southern municipalities of Nuevo Leon (Figure 1), the provision of water services to rural communities is less than 50%, and low percentages of sanitation and sewage services are notable. According to the Sociodemographic Overview of Mexico [53], the availability of piped water in Mier y Noriega is reported at a mere 28.9% of households. Similarly, the rest of the municipalities displayed low household access rates: Zaragoza (37.9%), Doctor Arroyo (39.9%), Galena (40%), and Aramberri (48.6%).

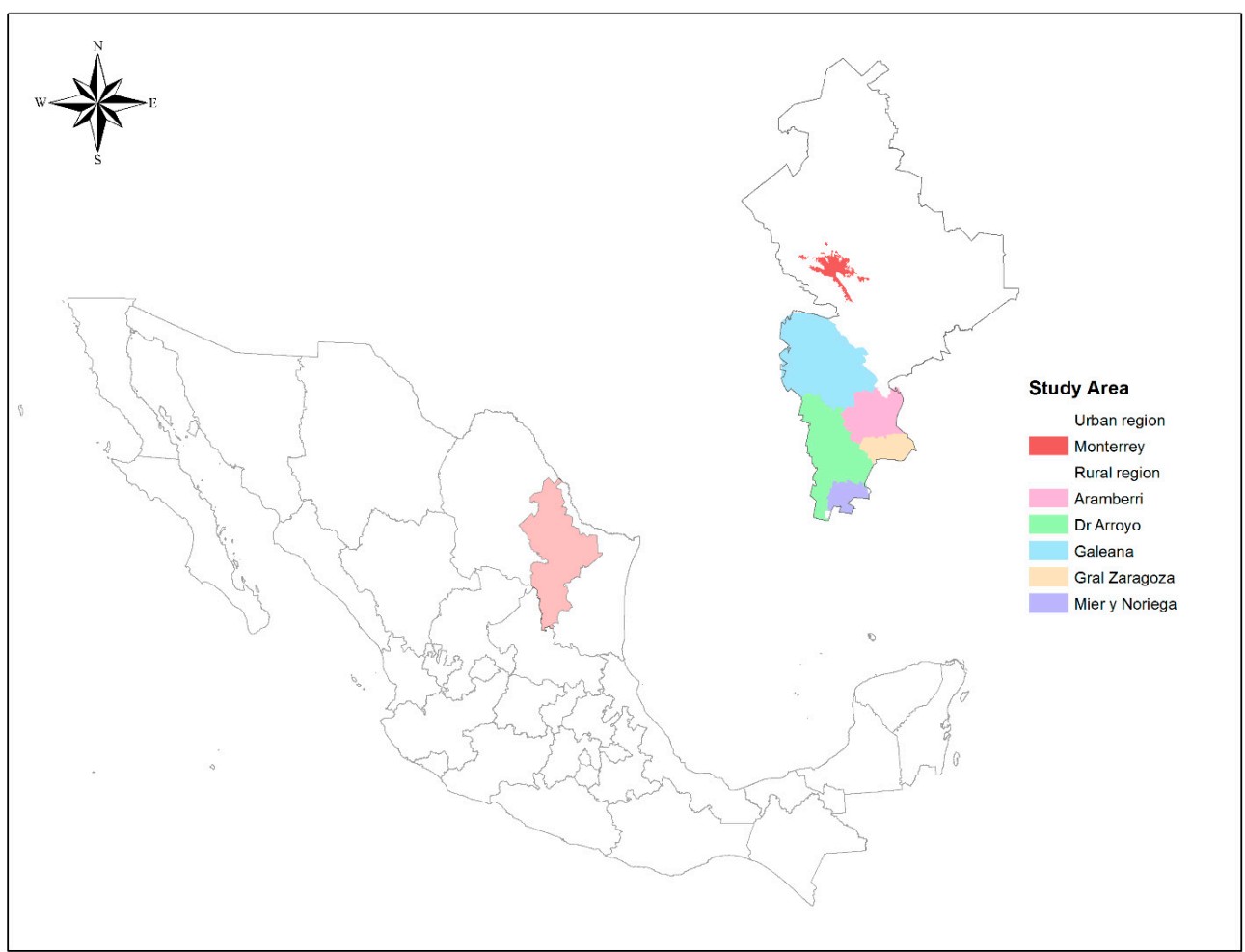

**Figure 1.** Geographic location of the study area. Source: Elaborated by the authors with data from the National Institute of Statistics and Geography [31].

*2.2. WPI*

Although water scarcity is a global problem, determining whether water is truly scarce is extremely complex due to climate change and environmental conditions; this aspect includes identifying whether it is a supply problem or a result of constantly increasing demands [15]. The WPI was developed to measure water poverty by assessing the relationship between socioeconomic conditions and water scarcity [55]. It provides an indirect method

for measuring a given quantity or state. This index method can measure water scarcity and the socioeconomic implications of a particular region [49] to enable a comparison of different areas or regions and replication for analyses of changes over time [56].

The WPI permits the numerical expression of water poverty on any scale based on physical, environmental, and socioeconomic parameters [57–59]. Although the WPI provides a numerical value on a scale from 0 to 1 to measure water poverty, the most valuable insights for understanding a particular situation of water poverty come from examining the individual factors or components that make up the index, rather than just relying on the overall index value [57].

The index calculation comprises five components [55]: Resources, Use, Access, Capacity, and Environment. Resources are defined as the physical availability of surface and groundwater. Access to water pertains to the level of approach to safe water for human use. Capacity in water management refers to the efficiency of the population in managing water use. Alternatively, Use denotes the manner in which a population uses water for diverse purposes. The environment component endeavors to capture environmental indicators that reflect the supply and management of water.

The general equation for calculating the WPI is derived as follows:

$$Water\ Poverty\ Index = \frac{\sum_{i=1}^{n} W_{X_i} Xi}{\sum_{i=1}^{n} W_{X_i}}. \tag{1}$$

According to Sullivan [57], the WPI for a particular region is the result of the weighted sum of *I* indicators involved in its calculation. The components are weighted according to their relative importance. Using weight functions $W_i$, weight *W* is applied to each component ($X_i$) of the WPI structure of a region, and $X_i$ is the value of each component. Equation (2) presents the development as follows:

$$Water\ Poverty\ Index = \frac{W_r R + W_r A + W_r C + W_r U + W_r E}{W_r + W_a + W_c + W_u + W_e}. \tag{2}$$

The WPI was developed to express the complex relationship between sustainable water resource management and poverty at all levels. The WPI can be calculated at not only the national level, but also the regional, district, or community level. The selection of indicators met the following requirements: indicators must be measurable and the necessary data must be readily available [50]. The reason is that the primary analysis can be used as a baseline to monitor the progress of water management projects.

### 2.3. Data Collection

As the WPI is an indicator-based approach, the first step included defining a set of indicators for its five components. The following criteria were used for the data selection and collection of indicators.

### 2.3.1. Resource (R)

The resource component was evaluated by considering the information on the annual production in water extraction wells for the supply of human consumption in each municipality. This investigation was based on a measure of the total availability of natural renewable water resources per inhabitant per day ($m^3$/inhabitant/day) and the reliability of the supply. The data used in this section were provided by the local agency in charge of the water supply in Nuevo Leon, (Agua y Drenaje de Monterrey IPD). Data on water resources were obtained from the website of the National Information System on Water Quality, Quantity, Uses, and Conservation (SINA) of the National Water Commission of Mexico.

### 2.3.2. Access (A)

The study selected three indicators for Access, namely, access to safe water (measured as the percentage of the population with access to improved and safely managed water sources (public service)); access to sanitation (calculated as the percentage of the population with access to sanitary drainage), and easy access to water (measured by the percentage of people that inhabits houses with water infrastructure). Data were collected from the Census of Population and Housing [12].

### 2.3.3. Capacity (C)

Capacity involves four socioeconomic indicators consulted from various sources. Data on infant mortality rates (C1) were derived from the Municipal Human Development Index of 2015 [60]. Based on the INEGI population and housing census [7], data on the percentage of the employed population with incomes of up to two minimum wages (C2) and the percentage of the population in houses without piped water (C3) were used as the capacity indicators of water scarcity. Data on the percentage of the population living in poverty (C4) were obtained from the Poverty Assessment by the National Council for the Evaluation of Social Development Policy [52]. These parameters were weighted to obtain the final value for Capacity.

### 2.3.4. Use (U)

Use was evaluated based on the use of water for different activities in all sectors (including domestic, agricultural, and industrial) and the economic impact of each use. In the MMA, 87% of the available water resources are used for human consumption. Meanwhile, in rural communities, only 4.4% of the water is used to meet the demand for human consumption. Data on the percentage of water used for human consumption per municipality (U1) were obtained from the documentation provided by the local agency in charge of water supply in Nuevo Leon (Agua y Drenaje de Monterrey IPD).

### 2.3.5. Environment (E)

This component highlights the environmental condition and integrity of water and ecosystem goods and services. This study used the rate of change in vegetation cover between 2020 and 2015 (E1) to describe the infiltration capacity of the ecosystems in the study area. Data obtained from satellite images of the normalized difference vegetation index (NDVI) of the two dates were used to identify the rate of changes in vegetation cover.

The NDVI, one of the most common indicators in land use assessments, quantifies vegetation cover using data collected from satellite imaging [47,48,61]. Based on how the plant absorbs and reflects light, this index measures the difference between two spectral bands, near-infrared (reflected by vegetation) and red light (absorbed by vegetation). The use of these two bands' renders highlights the degree of vegetative cover. This index also enables the discrimination of other elements in the satellite image, such as soil and water [51].

The NDVI was calculated using the difference between bands 4 and 5 of the Landsat 8 satellite; its range of values is limited between −1 and 1, where −1 indicates an area devoid of vegetation or bare soil, and positive values close to 1 indicate larger vegetation cover. Calculations for the NDVI were generated using a Geographical Information System (GIS), with the Image Analysis Tool in ArcGIS [47,62].

Satellite imaging data were downloaded from the Earth Observing System Data and Information System of the National Aeronautics and Space Administration of the United States.

Table 1 lists the indicators used to calculate the WPI. Appendix A presents all the data collected in this section.

**Table 1.** Components of the Water Poverty Index (WPI).

| Component | Definition | Indicator | |
|---|---|---|---|
| Resource | Physical water availability for human consumption | R1 | Availability of water per inhabitant per day |
| Access | Capacity of the population to access water for human consumption | A1 | Percentage of the population with water public service |
| | | A2 | Percentage of the population with access to sanitary drainage |
| | | A3 | Percentage of people who live in houses with water infrastructure |
| Capacity | Effectiveness of people's ability to manage water | C1 | Infant mortality rate |
| | | C2 | Percentage of employed population with an income of up to two minimum wages |
| | | C3 | Percentage of the population living in houses without piped water |
| | | C4 | Percentage of the population living in poverty |
| Use | Ways in which water is used for different purposes | U1 | Percentage of domestic water uses |
| | | U2 | Percentage of agricultural water uses |
| Environment | Relationship between water use and environmental conditions | E1 | Analysis of vegetation cover using NDVI |
| | | E2 | Surface affected by soil degradation |

Source: Elaborated by the authors on the basis of [59].

### 2.4. Data Analysis

After compiling the dataset, the study then conducted a statistical treatment of the data before calculating the WPI. Toward this end, a principal component analysis (PCA) and normalization of the variables were performed.

Indicators are expressed in various statistical units; thus, they must be normalized using a standard scale that renders the indicators comparable. Data normalization was performed using the min–max method. Each data series was left with a score between 0 and 1, where 1 represents the value that leads to higher water poverty and 0 denotes lower water poverty. Equation (3) was used to normalize the data.

$$X_i^* = \frac{X_{max} - X_i}{X_{max} - X_{min}} \tag{3}$$

Statistical Tests of the Dataset

The study conducted statistical examinations to assess the relationships between the variables. In cases where the variables exhibited significant correlations, they were excluded to eliminate redundancy. This study used PCA to analyze the dataset. Appendix A presents the outcomes of the statistical tests.

Based on the results of Pearson's correlation matrix, variables A3 and C2 were excluded from assessment. A3 was disregarded due to its limited representation among the access-related variables, primarily because it measures wastewater disposal instead of access to water. Meanwhile, C2 was excluded to prevent collinearity with variables C1 and C4, which exhibit high correlations.

### 2.5. Construction of WPI

The WPI calculation process is characterized by simplicity, cost effectiveness, and ease of comprehension. However, it has faced significant criticism [13,49,50]. Studies have identified two primary limitations of this index [49]. First, conducting WPI calculations

based on available data is typically challenging, because most estimates are tailored to fit the required data. Second, most WPI calculations assign equal weights to its components without providing any justification for this approach.

In response to these drawbacks, the current study proposes improvements. It addresses the challenge of performing WPI calculations using the available data by applying it to the municipality scale, because datasets related to demographics, economic activities, or infrastructure are frequently collected at the municipal level. Data on vegetation cover were calculated using remote sensing data, which offers advantages in filling data gaps for various applications, including environmental assessment and monitoring [48].

The second enhancement involves the application of multivariate techniques to objectively determine variable weights, providing an alternative to the common practice in WPI calculations of assigning equal weights to its components [49,62]. Therefore, this study explores three alternatives regarding the contributions (weights) of indicators to each component (subindex level):

Equal weights. All subindices exhibit the same weight. This alternative is used in the first WPI proposal [57] and is based on the idea that WPI calculation can be applied by a nontechnical audience in a simple and transparent manner.

Principal Component Analysis. PCA is used to allocate weights to each component. Multivariate methods, such as PCA, have proved to be more objective options for the weight assignment of WPI components [50]. The calculated weights are then normalized to ensure that the sum of the weights is equal to 1.

Analytic hierarchy process (AHP). AHP was implemented to propose each component weight as a new alternative that is particularly suitable for multicriteria decision making.

These weights were then assigned to the WPI components, and the index was calculated using Equation (2). Table 2 presents the final data used to construct the WPI, and Appendix B provides all the data calculations.

**Table 2.** Normalized data of each variable of the components for the WPI calculation.

| Locality | Components | | | | |
|---|---|---|---|---|---|
| | Resource | Access | Capacity | Use | Environment |
| Aramberri | 0.5698 | 0.7913 | 0.4908 | 0.0613 | 0.1068 |
| Dr. Arroyo | 0.9153 | 0.4251 | 0.5515 | 0.0572 | 0.0601 |
| Galeana | 0.4002 | 0.6393 | 0.5723 | 0.0404 | 0.1534 |
| Gral. Zaragoza | 0.0749 | 0.5396 | 0.6976 | 0.4697 | 0.0969 |
| Mier y Noriega | 0.4634 | 0.6593 | 0.4853 | 0.0222 | 0.1841 |
| South region | 0.2212 | 0.6320 | 0.5595 | 0.0392 | 0.1870 |
| MMA | 0.4183 | 0.1370 | 0.4960 | 0.7647 | 0.2113 |

## 3. Results

The results showed that, in both communities, rural and urban, water poverty exists. Water poverty values ranged from 0.5 to 0.68. Results closer to 1 indicate severe water poverty. Across all three weighting methodologies, rural communities exhibited higher levels of water poverty than the urban population. Access (A) and capacity (C) were the socioeconomic components that influenced the WPI results the most.

For geographical reference, Figure 2 displays the WPI results for the MMA, the urban population, and the municipalities in the southern part of Nuevo Leon, which represents the rural population. The outcomes of the three distinct methods for assigning weights to the components of the index are depicted.

Table 3 presents the calculated weights. ANOVA confirmed that the values of the different weighting techniques did not differ significantly ($p$-value < 0.05). The variables met the criterion of independence of the weights of the WPI components. The AHP was used as the third method to calculate the weight of each criterion or components of the WPI. The authors carried out the research by defining the problem, structuring the decision hierarchy, constructing matrices, and calculating the relative weights using bibliographical

data. Subsequently, the authors' decisions underwent thorough review and validation by both experts and the local water supply agency in Nuevo Leon, known as "Agua y Drenaje de Monterrey IPD".

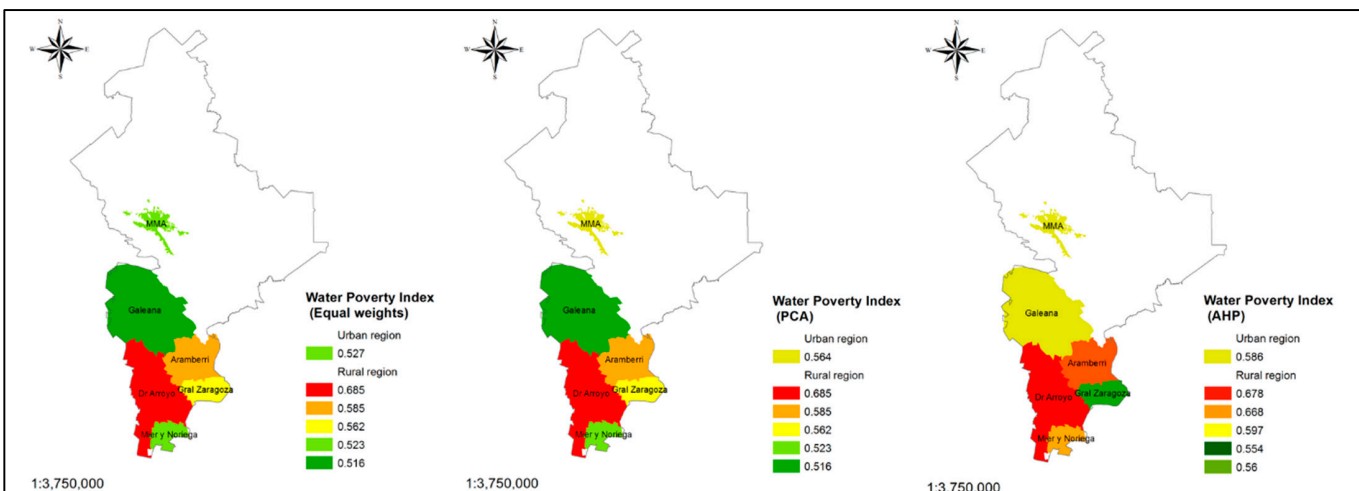

**Figure 2.** WPI map illustrating variations among the three weighting methods: equal weights, principal component analysis (PCA), and analytic hierarchy process (AHP), along with distinctions in water poverty between urban and rural areas.

**Table 3.** Component weights using the three methodologies for the WPI calculation.

| Component | Weights | | |
|---|---|---|---|
| | Equal Weights | PCA Weights | AHP |
| Resource | 0.2 | 0.185 | 0.158 |
| Access | 0.2 | 0.152 | 0.319 |
| Capacity | 0.2 | 0.275 | 0.212 |
| Use | 0.2 | 0.222 | 0.221 |
| Environment | 0.2 | 0.166 | 0.09 |

The index was calculated at various scales, such as using the municipal and regional scales, the data mean of all the southern municipalities of Nuevo Leon as the rural region, and the MMA as the urban region. The WPI was calculated for each location using the three weighting methodologies. Table 4 presents the results. A ranking result close to 1 indicates the locality with the highest WPI and thus the poorest in water. According to the results, the water poverty values ranged from 0.5 to 0.68. In each of the three weighting methodologies, the results demonstrated higher levels of water poverty in the rural communities of southern Nuevo Leon than those in the urban population of Monterrey.

**Table 4.** Calculated values of the WPI.

| | WPI | | |
|---|---|---|---|
| Locality | Equal Weights | PCA | AHP |
| Aramberri | 0.597 | 0.585 | 0.668 |
| Dr. Arroyo | 0.661 | 0.685 | 0.678 |
| Galeana | 0.510 | 0.516 | 0.560 |
| Gral. Zaragoza | 0.539 | 0.562 | 0.554 |
| Mier y Noriega | 0.506 | 0.523 | 0.597 |
| South Nuevo Leon | 0.563 | 0.574 | 0.612 |
| MMA | 0.527 | 0.564 | 0.586 |

Figures 3 and 4 depict the results of the WPI calculations using pentagrams. This graphical representation demonstrates which components primarily contribute to water poverty and presents the estimations using the three weighting methods. The comparison of the contributions of each component extends to urban and rural contexts (Figure 3) and among the municipalities situated in the southern region (Figure 4).

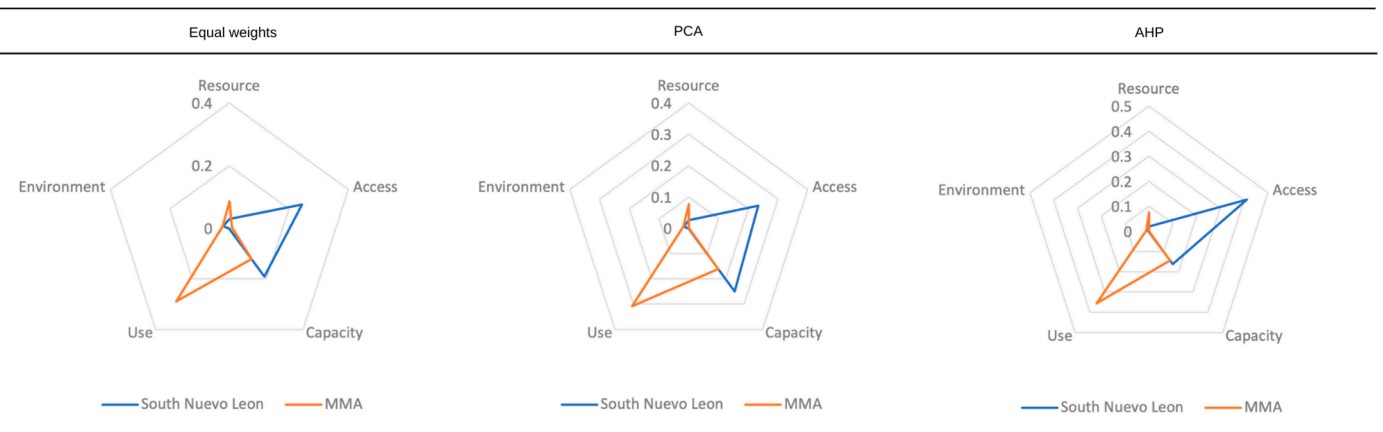

**Figure 3.** Components of the final WPI in the urban and rural contexts, presenting the score for each component and comparing the three weighting methods (equal, PCA, and AHP).

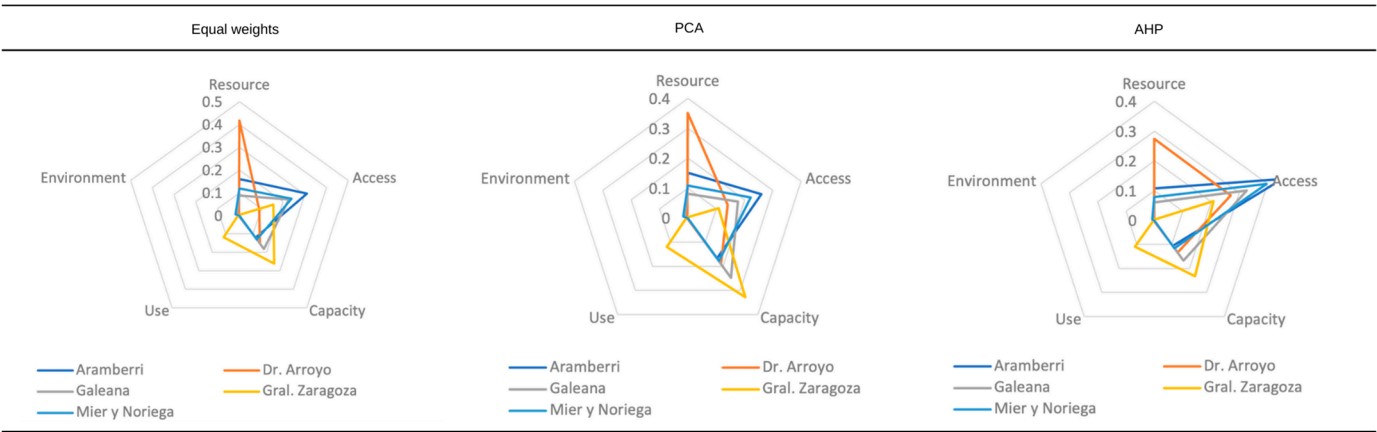

**Figure 4.** Components of the final WPI among rural municipalities, showing the score for each component and comparing the three weighting methods (equal weights, PCA, and AHP).

## 4. Discussion and Conclusions

The findings of the WPI analysis indicated that water poverty exerted an impact on every community within the study area, with scores above 0.5. This disparity was expected because, although the territories share similar geographic and climatic conditions, they differ in terms of political capacity and resources. This aspect is reflected in various administrative advantages that combine to enable the MMA to continue its demographic and economic growth. Rural communities in southern Nuevo Leon exhibited higher levels of water poverty than their counterparts in the urban area. These findings are consistent with those of previous studies conducted in the same areas [17,24,25,53,63], which also emphasized the urgent and critical water shortage that affects sustainable development in the region. A further examination of this subject will be presented later in this section.

### 4.1. WPI Performance

The index calculation faced two challenges while addressing the limitations of the WPI. One challenge was related to the availability and scale of data, whereas the second

pertained to the methods for weighting the components of the index [49,50]. To address the first challenge, this analysis involved a meticulous data processing step to avoid cross-subsidization between regions using GIS; these analyses were adapted from previous references [37,47,58]. Data from different scales were collected for this analysis, including socioeconomic data (collected at the municipal level) and physical water data (available at the watershed, or state level). The resulting data underwent statistical tests to refine the information of the selected indicators used in the calculations.

Second, the study employed different weighting methods to assess the contribution of each component to water poverty. The results displayed variations in the water poverty rankings among the municipalities based on the weighting methods. However, similar rates of water poverty were estimated across municipalities and populations, regardless of the weighting methodology used.

An examination of the three weighting methods for index calculation consistently shows strong or similar rates of the municipalities experiencing the most severe poverty. In the equal weighing of each component method, the municipality with the highest level of water poverty was Dr. Arroyo, which agreed with previous studies, followed by Aramberri [36,54,63]. In the PCA weighting method, Dr. Arroyo, once again, exhibited the poorest water conditions. However, the resulting indices using AHP identified the indicators that contribute to the high rates of water poverty. The study also determined that these attributes are closely and primarily linked in the rural municipalities of Dr. Arroyo and Aramberri.

However, a nuanced picture emerges when examining the less critical areas, where the methodologies exhibit variations. In the equal weighing method, the WPI results demonstrated that rural communities in Mier y Noriega followed by Galeana were the least water poor of the municipalities. In the PCA approach that was used to assign the contribution of each component to poverty, Galeana was the community that was least affected by water poverty, followed by Mier y Noriega. In these two WPI calculations, the water poverty estimations for the Mier y Noriega municipality were not expected. As indicated by previous studies, the rural population living in the municipalities of Doctor Arroyo and Mier y Noriega are the most affected by the water scarcity problem in Nuevo Leon [6,8].

These findings suggest a potential underestimation for certain areas when employing equal weights and when using PCA in some cases. The relatively small population size of Mier y Noriega may account for the underestimation of water poverty. The index seemingly prioritizes water supply initiatives in larger, more densely populated communities. Thus, pointing out that underestimation due to low populations compared to other municipalities can be a critical concern is essential when drawing attention to marginalized communities [45,50].

For this analysis, AHP demonstrated a better performance, which aligns with the existing literature. However, integrating documentary information into these types of evaluations remains crucial. This consideration is vital, because statistical data alone may inadvertently divert attention from the specific needs of certain minorities and marginalized communities [63]. In Mier y Noriega, more than 70% of the population lives in scattered towns inhabited by a few people; out of them, the majority may be composed of one or two families [24,36]. This context explains the lack of basic services and marginalization experienced by these communities [29]. For cases of underestimation similar to this, other authors have argued that the WPI helps to elevate the debate on the concept of water poverty and that the index may be better suited to political arguments rather than statistical measurements [64].

Nonetheless, the weighed components resulting from the AHP exhibited results for the water poverty of these municipalities that agree with the estimations argued by other studies and experts [8,24,25,36,48,52,65]. The WPIs calculated on the basis of AHP multicriteria decision making indicated that Dr. Arroyo and Aramberri were the most vulnerable municipalities in terms of water poverty. Conversely, Gral. Zaragoza and

Galeana appeared to be the least affected by water poverty. The findings for municipalities with the least and most water poverty correspond to those in the referenced literature, which implies an accurate assessment of water poverty using this methodology.

Given that the most valuable information lies in the components instead of in the final single number [56], this evaluation identified differences between the variables that influence water poverty across populations. However, acknowledging that the significance of individual components may change over time is crucial [45]. This temporal aspect not only offers an advantage in monitoring water scarcity and access, but also emphasizes the essential need for periodic updates. These updates are imperative for maintaining the accuracy and alignment of assessments with the evolving dynamics of water-related challenges across populations [51].

The WPI component scores (Table 4) identified the underlying causes of water poverty and their variations within the municipalities in the study area and among the urban population. Specifically, the primary contributing factor to water poverty in rural communities is water access. These communities struggle with constraints in accessing water and meeting the human demands for water. Conversely, the urban population experiences water poverty with a different challenge. Specifically, the difficulty faced by people in the MMA is mainly due to deficient water management and insufficient water resources to meet the city's population demand.

Notably, rural communities in southern Nuevo Leon are facing significant water poverty and require urgent action to increase access to safe drinking water and enhance the capacity to manage water. However, a consistent finding indicated by prior studies and reaffirmed by our research efforts through fieldwork and observations in the study area [26,65] is that rural communities in southern Nuevo Leon exhibit lower levels of dependence on water resources than those of the city, despite this scarcity. In this region, the predominant factors contributing to the elevated levels of water poverty are a result of profound deficiencies in water management and administration [10,18].

Based on the data and bibliography, rural communities displayed a notable efficiency in using water for human consumption, because most water is allocated for agricultural purposes [54]. Although these communities displayed notable skills in water management for domestic use, the analysis highlights a strong need to enhance access to clean water, followed closely by strengthening water capabilities. These primary factors that drive water poverty, as indicated by the PCA and AHP, are supported by the control results derived from the equal weighting method of component calculations used in the WPI. This finding aligns with those of other studies [41,62], which highlight the improvement in employing the PCA and AHP weighting methods to identify the components responsible for water poverty.

Nonetheless, the urban population in the MMA is also facing challenges with water poverty. This struggle stems from a combination of high and escalating water demand, lack of effective water management, and sustainable urban development practices. The reason is that the high level of water consumption is mainly for domestic use, followed by the water footprint of the industry and energy sector [11,25,41].

The MMA has access to various sources of water, including surface and groundwater. Collectively, these sources contribute to the water supply of the city. However, the high water demand, which is linked to the growing urban population and industrial activities, has strained these water resources [8]. Remarkably, the city managed to meet this heightened demand until 2020 due to substantial investments in hydraulic infrastructure. This increased demand for water along with the arid and semidesert conditions and recurrent severe droughts in the region have exacerbated the challenges associated with water scarcity [66].

The MMA poses advantages in climate and geographical location compared to the southern rural areas because the city is near the Cumbres National Park (protected area), which is the main water catchment and infiltration area of water in Nuevo Leon [8,25]. However, deficient management along with corruption has led to the degradation and

fragmentation of the ecosystems within the national park [24]. This poses a threat to the conservation of natural vegetation areas that contribute to water catchment and filtration. Estimates for the MMA indicate that the watershed catchment that supplies the city is insufficient for meeting the projected needs in the coming decades [9,24,25,54,66]. These estimations highlight the importance of conserving natural areas for water capture and infiltration and implementing urgent urban development measures.

However, scarcity is not the only problem in Monterrey. It currently faces the issue of polluted surface and groundwater due to industrial discharge and domestic waste [8,11,54]. Evidently, this scenario poses a damaging effect on the water available for human consumption.

In the case of the rural communities of southern Nuevo Leon, the primary use of water is agricultural production. Water extraction for this economic sector has diminished the water availability of aquifers. The remaining water may be polluted by salts and substances associated with fertilizers and pesticides [63]. This assessment could be more informative if the water quality accounted as a sixth component of the WPI, as suggested by other regional studies [44]. However, the current study did not consider this particular component, because data had to be collected on-site and subsequently analyzed using water quality tests in the laboratory. This study aimed to develop a methodology that could be implemented broadly using pre-existing and easily obtainable data, facilitating its further implementation in other rural communities within the arid regions of northern Mexico. This approach ensures that water poverty assessments can be conducted without the need for substantial financial resources.

*4.2. Disparities in Water Scarcity*

Recognizing that water poverty is a multidimensional challenge with interconnected variables is essential. Despite the region analyzed being located in a semidesert ecosystem with low precipitation and vulnerability to severe droughts, other variables demonstrated a higher impact on water poverty than water availability as a result of environmental conditions. The current findings indicate that differences in access to water, capacity to manage water, and water use can contribute to different types of water poverty and have diverse impacts on socioeconomic conditions. Low socioeconomic conditions directly impact the ability to access safe drinking water [26,28,34]. Communities with low-income levels, weak health, and poor educational systems are expected to have inhabitants lacking access to safe drinking water [38]. Water scarcity engenders a host of socioeconomic and environmental issues, which create a feedback loop that perpetuates scarcity. This circular phenomenon underscores the complexity of the challenge and necessitates comprehensive approaches that address its multifaceted nature (Figure 5).

However, the most contrasting conditions between the southern rural communities and the urban population of Nuevo Leon are found in the socioeconomic dimension. Monterrey is highly populated with continued housing and economic growth, which translates into accelerated urbanization. Consequently, building extensive hydraulic infrastructure to meet the growing needs of the population and the economy to meet the increasing demand for water supply is an essential aspect [27,66]. Resources and efforts in Nuevo Leon have been predominantly directed toward providing water to the urban population by investing in hydraulic infrastructure that has nearly reached 100% coverage of both water supply and sanitation [11]. Despite these efforts, the demand for water in the MMA does not cease [8,9].

Government budgets are often limited, and the high cost associated with water supply in rural areas is often used to justify why efforts and investments related to water management have been primarily channeled toward urban areas [19,42]. The MMA is considered to be more profitable for public investment because of its higher population density and the presence of economic activities. Infrastructure investments can serve a larger number of people and ensure economic growth and development in the city; thus, providing water services in the city is more affordable and accessible [25,41]. Conversely, providing water services to remote areas inhabited by rural populations is more expensive

in terms of government investment in infrastructure. Focus on supplying water to the city has exerted a negative consequence by diverting attention away from the needs of rural communities, which potentially leads to the water-related challenges or deficiencies they face [26,34].

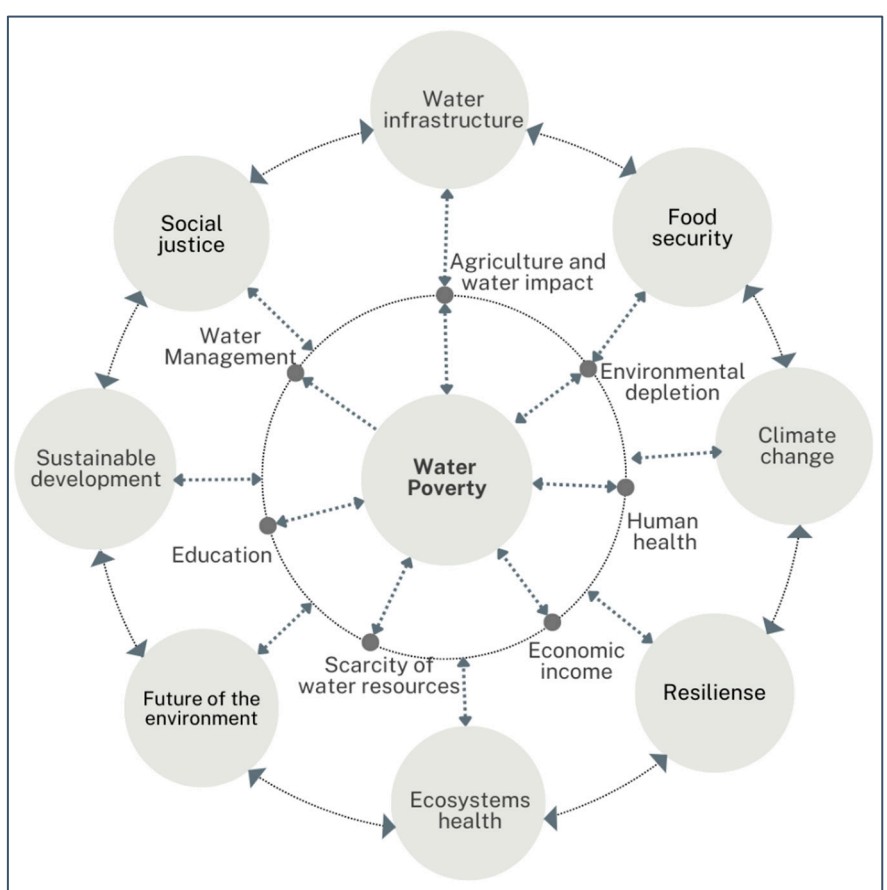

**Figure 5.** Cycle of water poverty.

### 4.3. Sustainable Development

As demand for water increases and its availability diminishes due to climate change, the perspective of prioritizing the water needs of the urban population has posed a threat to the implementation of water management strategies that target improvements in water poverty conditions in marginalized communities [26,29]. Rural areas frequently serve as the main source of water [28,34]. This aspect raises concerns about equitable resource allocation and the failure to adequately protect and manage watershed areas, which are essential for maintaining a sustainable and sufficient supply of water.

Consequently, effects on the environment, such as the overextraction of water, impact social development in rural communities. Moreover, this social inequality intensifies when rural areas face marginalization. Ultimately, this cycle of water scarcity perpetuates poverty in rural communities, which, in turn, fosters migration to cities, and further hampers water availability [15,34,43].

Water poverty is linked to economic disparities that can trigger migration to cities, which contributes to overcrowding and competition for resources. The poverty situation in the rural population can lead to migration and the abandonment of agricultural activities [60]. Monterrey is susceptible to more immigration flows that will increase the population's water demand, which impacts the socioeconomic situation and increasingly complicates the achievement of the SDGs [35,43].

The development in the study area is far from sustainable. This study identified key indicators for alleviating the water crisis in the region. The findings also suggest the potential

for sustainable development. A focus on sustainable development involves taking prompt action to address socioeconomic disparities between rural and urban populations [66]. In this sense, the MMA can promote efficient and climate-related management of water resources to ensure that such resources are available to meet the needs of its population. As suggested by other studies [8,11,24,46], this study advocates the urgent adoption of enhanced management strategies that will impact urban and rural communities, thereby reducing the pressure on the water resources in the region.

Furthermore, the dependency of rural and urban communities on groundwater should be reduced to meet future water demands [10]. Thus, conservation efforts in the water basin wherein the MMA is located are necessary. Groundwater sustains economic activities and the well-being of the population. Thus, the future of the city depends on the conservation of watershed ecosystems. Ecological integrity is imperative for enhancing environmental conditions across the region.

In Monterrey, water poverty may be alleviated if the population were to reduce water demand by enhancing water use toward sustainable management [41]. The water resilience capacity in the city can be improved by reducing and controlling water utilization and increasing the availability of clean water by eliminating pollutants discharged by industries [11,66]. Awareness of the population regarding water scarcity is also crucial for better house water use practices [11,41].

However, new supply sources will be required in the long term to meet the demand for water resources in Monterrey, which will require more effective management of demand and current sources [8,11,66]. Improvements would then be needed in the well-being of rural populations, because efforts to conserve natural resources, including water, which is a vital resource of any human activity, can be made in rural communities [10,29]. Thus, the city is dependent on the people inhabiting rural regions to perform adequate water management and utilization to meet the SDGs.

Addressing water scarcity in rural municipalities requires the urgent promotion of and focus on management strategies that aim to enhance water infiltration and recharge aquifers [20,31,67]. This measure involves supporting initiatives such as rainwater harvest and collection [26,68,69]. Although they are crucial for alleviating acute water scarcity, the successful implementation of these measures in rural communities across the southern region of Nuevo Leon is heavily reliant on effective community participation that plagues the region [24,28].

Community engagement plays a pivotal role in designing and implementing sustainable solutions for alleviating water scarcity [28,33,67]. Despite the initial adoption of water-harvesting technologies and sustainable water management practices in these rural communities, the long-term success of technology transfer and implementation faces persistent challenges. Factors such as changing community priorities; insufficient resources or support; and unforeseen socioeconomic, cultural, or political barriers have hindered communities from sustaining the implementation of these strategies [26,35]. Consequently, the involvement of local inhabitants is crucial for effective water management practices [34,67,70]. Empowering communities toward water management can foster collaborative work and ensure the long-term integrity of natural resources and human well-being.

## 5. Conclusions

Climate change is expected to affect water availability by reducing precipitation in arid regions. Thus, an urgent need emerges to improve the water poverty situation to realize the SGDs and sustainable development in developing countries. This study selected the WPI to assess water poverty and contribute to informed, prudent, and responsible decisions on water management and socioeconomic development policies that can strengthen strategies for poverty alleviation and environmental conservation.

The results indicated disparities between rural and urban populations. Specifically, the results of the three weighting methods revealed that rural communities exhibited higher levels of water poverty compared to the urban population. Despite the consistency

of the WPI scores across the three methods, the modified WPI presented in this study demonstrated the importance of considering weights. In the standard equal weight of all components, we observed an underestimation in the case of a small population that was not statistically representative. The informed AHP demonstrated a better performance, which aligned with the existing literature and highlighted the importance of integrating documentary information into statistical data.

This evaluation identified the contribution of the variables that influence water poverty in each population. Water poverty in rural communities is related to access to safe drinking water and the social capacity to manage water compared with the MMA, in which the most urgent attention should be given to water use and environmental integrity. These results intend to prevent diagnosis from obscuring the reality of water on a local scale when statistics may underestimate the situation of regional water scarcity.

The findings indicate the necessity of holistic and sustainable approaches to addressing water poverty in rural and urban settings. Actions toward reductions in socioeconomic disparities are the key to improving the current water crisis. This focus on the urban population has implications for sustainable development. Water resilience in the MMA can be enhanced through efficient water use practices, pollution control, and public awareness. Mitigating water scarcity in rural communities includes reducing dependence on groundwater and promoting conservation efforts in watershed ecosystems, with an emphasis on community well-being and participation. Therefore, empowering communities in water management is important for ensuring the long-term integrity of natural resources and sustainable development.

The data obtained from the study can inform further decisions on improving the water poverty situation made by policymakers and water planners. The relevance of the modified WPI lies in its broad applicability to various stakeholders. Its operational simplicity allows for calculation while also enabling statistical and spatial analyses. Planning water management would be relatively straightforward if the relevant strategies were related to the needs of the local socioeconomic and ecological conditions. Future studies should explore the application of the WPI methodology at different regional levels and consider the three weighting methods. Comparing the results using the different methods can enhance the coherence and interpretability of the WPI values.

Assessments of water poverty continue to present a formidable challenge. This methodological framework has the potential for a broader application to other regional levels based on the data available for Mexican municipalities. This research can offer insights for subsequent studies that focus on marginalized groups and WPI implementation to draw a larger picture of the relationships between the indicators of water scarcity and the socioeconomic conditions of a particular region.

**Author Contributions:** Conceptualization S.P.-T.; methodology, S.P.-T.; validation, S.P.-T. and M.G.M.-C.; formal analysis S.P.-T. and M.G.M.-C.; writing original draft preparation S.P.-T.; supervision M.G.M.-C. All authors have read and agreed to the published version of the manuscript.

**Funding:** We sincerely thank the Mexican Council for Humanities, Sciences, and Technologies (CONAHCYT) for granting a Ph.D. scholarship to Silvana Pacheco-Treviño (690358). This support significantly contributed to the successful completion of this research.

**Institutional Review Board Statement:** Not applicable.

**Informed Consent Statement:** Not applicable.

**Data Availability Statement:** The data presented in this study are available on request from the corresponding author.

**Acknowledgments:** We acknowledge the ongoing support from Tecnologico de Monterrey, which has played a vital role in the advancement of our work. We are grateful for the resources and opportunities provided by both CONAHCYT and Tecnologico de Monterrey that have enriched the quality of this study.

**Conflicts of Interest:** The authors declare no conflicts of interest.

## Appendix A. Data Collection and Statistical Tests

The data collection encompasses the diverse components of the WPI, including Resources (as explained in Section 2.3.1), Access (outlined in Section 2.3.2), Capacity (explained in Section 2.3.3), Use (described in Section 2.3.4), and Environment (elaborated on in Section 2.3.5).

## Appendix B. Data Calculations and WPI Construction

This includes detailed statistical analyses, such as principal component analysis (PCA) and normalization procedures, outlined in Section 2.4. And the application of multivariate techniques for determining variable weights, as detailed in Section 2.5.

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
