# Peer review of "The Socioeconomic Dimensions of Water Scarcity in Urban and Rural Mexico: A Comprehensive Assessment of Sustainable Development"

_sustainability, doi:10.3390/su16031011_

Round 1

Reviewer 1 Report

Comments and Suggestions for Authors

This paper “Socioeconomic Dimensions of Water Scarcity in Urban and Rural Mexico: A Comprehensive Assessment of Sustainable Development” highlights significant contrasts in water scarcity between these populations, suggesting the need for customized solutions based on regional, territorial, and cultural characteristics. The study also identifies key indicators for alleviating the water crisis and suggests potential for sustainable management. There is a need for prompt action to improve water access, with a focus on addressing the disparities between rural and urban populations. The application of this paper is interesting, but those flaws should be fixed before a proper review. My comments are as follows.

The paper needs extensive editing of English language.

The structure of Introduction can review more papers about Water Scarcity monitoring, especially novel applications they have provided in SDG monitoring and assessment. For example, “Big Earth Data in Support of the Sustainable Development Goals” book provide a lot of examples in SDGs monitoring.

Line 193: “The author” should be the authors as there are several authors listed in the manuscript.

Line 296: same problem as above.

Line 354: The rank of colorbars should be identical in Figure 2. The rank of colorbar in Water Poverty Index (AHP) is ascending, but others are descending.

Line 377: The fonts in Figure 3 should be identical and should have the same dark color. The “a” and “b” should be described in Figure caption. Figure 3 should be divided into 3 different figures or “a” and “b” should not be iteratively presented in the bottom panels.

In my opinion, the discussion in this paper can be divided into several subsections to illustrate the main findings in this study.

Discussion and conclusions should be divided into two sections. The conclusion seems ok, but update the important finding obtain from above queries in the conclusion section.

Comments on the Quality of English Language

Extensive editing of English language required

Author Response

Response to Review of Manuscript ID Sustainability-2736140

We would like to express our gratitude to the reviewer for the insightful comments and valuable suggestions provided for the manuscript titled "Socioeconomic Dimensions of Water Scarcity in Urban and Rural Mexico: A Comprehensive Assessment of Sustainable Development". We have carefully revised the manuscript according to the Reviewer’s insightful comments and provided point-by-point responses in the following section.

Comment 1: This paper “Socioeconomic Dimensions of Water Scarcity in Urban and Rural Mexico: A Comprehensive Assessment of Sustainable Development” highlights significant contrasts in water scarcity between these populations, suggesting the need for customized solutions based on regional, territorial, and cultural characteristics. The study also identifies key indicators for alleviating the water crisis and suggests potential for sustainable management. There is a need for prompt action to improve water access, with a focus on addressing the disparities between rural and urban populations. The application of this paper is interesting, but those flaws should be fixed before a proper review. My comments are as follows.

Response 1: Thank you for your positive feedback on the overall content of the manuscript. We, however, made minor adjustments to each section, aiming at refining grammar and enhancing the overall cohesiveness of the fundamental premise of the manuscript.

Comment 2: The paper needs extensive editing of English language.

Response 2: We appreciate Reviewer 1 for highlighting the need for extensive editing of the English language in the manuscript. We have carefully revised the entire manuscript to improve language clarity, coherence, and grammatical accuracy. Professional editing services have been employed to address this concern effectively.

Comment 3: The structure of Introduction can review more papers about Water Scarcity monitoring, especially novel applications they have provided in SDG monitoring and assessment. For example, “Big Earth Data in Support of the Sustainable Development Goals” book provide a lot of examples in SDGs monitoring.

Response 3: We acknowledge the suggestion to enhance the Introduction section by incorporating a more comprehensive review of literature on Water Scarcity monitoring, especially focusing on novel applications in Sustainable Development Goals (SDG) monitoring. We have revised the Introduction to include relevant references regarding water scarcity evaluations in the region, examples of evaluations using the selected methodology, and also examples from Big Earth Data implementation for Sustainable Development Goals assessments."

Comment 4: Line 193: “The author” should be the authors as there are several authors listed in the manuscript. Line 296: same problem as above.

Response 4: We have corrected instances where "The author" appeared, ensuring proper acknowledgment of all contributing authors throughout the document.

Comment 5: Line 354: The rank of colorbars should be identical in Figure 2. The rank of colorbar in Water Poverty Index (AHP) is ascending, but others are descending.

Line 377: The fonts in Figure 3 should be identical and should have the same dark color. The “a” and “b” should be described in Figure caption. Figure 3 should be divided into 3 different figures or “a” and “b” should not be iteratively presented in the bottom panels.

Response 5: We appreciate the meticulous feedback on the presentation of figures. The issues with the colorbar in Figure 2 and the fonts in Figure 3 have been addressed. The colorbars in Figure 2 have been adjusted to ensure uniformity, and the fonts in Figure 3 are now identical with a consistent dark color. Additionally, Figure 3 has been restructured and divided into 2 figures to improve clarity. Also, figure captions have been updated accordingly.

Comment 6: In my opinion, the discussion in this paper can be divided into several subsections to illustrate the main findings in this study.

Response 6: We acknowledge the suggestion to divide the discussion into several subsections to better illustrate the main findings of the study. The discussion section has been reorganized into subsections to enhance the clarity of the main findings.

Comment 7: Discussion and conclusions should be divided into two sections. The conclusion seems ok, but update the important finding obtain from above queries in the conclusion section.

Response 7: We appreciate the recommendation to divide the discussion section into two parts. We included a conclusion section for clarity. This section has been revised accordingly, incorporating important findings from the queries raised during the review process.

Once again, we sincerely thank Reviewer 1 for the thoughtful feedback and constructive suggestions. These improvements contribute significantly to enhancing the overall quality and readability of the manuscript.

Best Regards,

Silvana Pacheco-Treviño and Mario Guadalupe Manzano-Camarillo
Corresponding Author: Silvana Pacheco-Treviño

Reviewer 2 Report

Comments and Suggestions for Authors

The article is used the water poverty index to provide valuable insights and assist stakeholders in developing comprehensive strategies for local conditions.

 The study employed different weighting methods, including equal weighting, PCA,

and AHP, to assess the contribution of each component to water poverty.   The study noted differences in water poverty rates between urban and rural populations.   Water poverty is linked to economic disparities that can trigger migration to cities, which contributes to overcrowding and competition for resources.   Across the three weighting methodologies rural communities exhibited higher levels of water poverty compared to the urban population.

There are some problems, which must be solved before it is considered for publication.   Frist, relevant research background needs to be supplemented in INTRODUCTION.  Second CONCLUSIONS needs more in it, as it's more of an afterthought. The authors are suggested to highlight important findings and include afterthought of this work.

Comments on the Quality of English Language

Minor editing of English language required

Author Response

Authors' Reply to Reviewer 2 Comments

We extend our appreciation to Reviewer 2 for the thoughtful review of our manuscript titled "Socioeconomic Dimensions of Water Scarcity in Urban and Rural Mexico: A Comprehensive Assessment of Sustainable Development," submitted to Sustainability. We have made an effort to address each of the reviewer’s comments in a structured manner, aiming to simplify the editorial review process.

Comment 1: The article is used the water poverty index to provide valuable insights and assist stakeholders in developing comprehensive strategies for local conditions. The study employed different weighting methods, including equal weighting, PCA, and AHP, to assess the contribution of each component to water poverty. The study noted differences in water poverty rates between urban and rural populations. Water poverty is linked to economic disparities that can trigger migration to cities, which contributes to overcrowding and competition for resources. Across the three weighting methodologies rural communities exhibited higher levels of water poverty compared to the urban population.

Response 1: We thank Reviewer 2 for recognizing the significance of our study in employing the Water Poverty Index (WPI) and various weighting methods, including equal weighting, Principal Component Analysis (PCA), and Analytical Hierarchy Process (AHP), to assess the contribution of each component to water poverty. Your acknowledgment of the observed differences in water poverty rates between urban and rural populations reinforces the importance of our findings.

Comment 2: There are some problems, which must be solved before it is considered for publication. Frist, relevant research background needs to be supplemented in INTRODUCTION. Second CONCLUSIONS needs more in it, as it's more of an afterthought. The authors are suggested to highlight important findings and include afterthought of this work.

Response 2: We appreciate the insightful comment on the linkage between water poverty and economic disparities leading to migration to cities, contributing to overcrowding and resource competition. The emphasis on the higher levels of water poverty in rural communities, as highlighted across the three weighting methodologies, is duly noted.

  1. Introduction Section: We acknowledge the suggestion to supplement relevant research background in the Introduction. The Introduction has been revised to provide a more comprehensive overview of existing research on water poverty, including its implications and the methodologies used.
  2. Conclusion Section: We appreciate the feedback on the need for additional content in the Conclusion section. The Conclusion has been expanded to include a more thorough discussion of important findings and their implications. This revision aims to offer a more robust and reflective afterthought of the work conducted.

We acknowledge the suggestion for minor editing of the English language. Professional editing services have been engaged to address language-related issues and enhance the overall quality of the manuscript.

We are grateful for the constructive feedback provided by Reviewer 2, and we believe that the suggested improvements have significantly strengthened the manuscript. Your comments have been instrumental in refining our work, and we look forward to the opportunity for further collaboration.

Best Regards,

Silvana Pacheco-Treviño and Mario Guadalupe Manzano-Camarillo

Corresponding Author: Silvana Pacheco-Treviño

Reviewer 3 Report

Comments and Suggestions for Authors

Great paper and concept. I have a few major comments and a few editorial items. 

1. The weighting factors are considerable parameter and this is mentioned but it is mentioned later on. I would suggest mentioning this when you introduce the WPI. At the least what the weighting factor are and how they are figured or estimated.

2. When the NDVI is mentioned  you might make a few references to it as all of your readers may not understand principle and its ubiquity in science. 

3.  When percentages are mentioned make sure it is clear what the percentage are of, in the paper when percentage are stated its not clear what the object (all water total, ground water, surface water, domestics...) of percentage is.

4. two-third should be plural - line 37

5. In the intro it is mentioned about water "availability" with usage rates - it is referenced, which is great, but what does it mean. Because the paper is about water scarcity it is important to stress what is "availability."

6. Finally, the abstract well describes the paper but it does not mention the key attributes of the WPI, the principle components or weighing factors, or the recommendations of the authors. I think this is mistake because that is the principal focus of the paper. 

Author Response

We would like to express our gratitude to the reviewer for the insightful comments and valuable suggestions provided for the manuscript titled "Socioeconomic Dimensions of Water Scarcity in Urban and Rural Mexico: A Comprehensive Assessment of Sustainable Development". We have carefully revised the manuscript according to the Reviewer’s insightful comments and provided point-by-point responses as follows:

Comment 1: The weighting factors are considerable parameter and this is mentioned but it is mentioned later on. I would suggest mentioning this when you introduce the WPI. At the least what the weighting factor are and how they are figured or estimated.

Response 1: We appreciate the suggestion to provide a more detailed introduction to the weighting factors when introducing the Water Poverty Index (WPI). The manuscript has been revised to include a clearer explanation of the weighting factors in the introduction as well as in the materials and methods sections, including how they are determined or estimated. This adjustment ensures a better understanding for readers from the outset.

Comment 2: When the NDVI is mentioned you might make a few references to it as all of your readers may not understand principle and its ubiquity in science.

Response 2: Thank you for pointing out the need to elaborate on NDVI (Normalized Difference Vegetation Index) and its significance. The manuscript has been updated to include additional references and explanations to help readers unfamiliar with NDVI understand its principles and its widespread application in scientific research.

Comment 3: When percentages are mentioned make sure it is clear what the percentage are of, in the paper when percentage are stated its not clear what the object (all water total, ground water, surface water, domestics...) of percentage is.

Response 3: We acknowledge the concern about clarity in stating percentages. The manuscript has been revised to explicitly define the context of percentages, ensuring transparency in the presentation of data

Comment 4: two-third should be plural - line 37

Response 4: The suggestion to correct the plural usage of "two-third" to "two-thirds" in line 37 has been duly noted and incorporated into the manuscript.

Comment 5: In the intro it is mentioned about water "availability" with usage rates - it is referenced, which is great, but what does it mean. Because the paper is about water scarcity it is important to stress what is "availability."

Response 5: We appreciate the comment regarding the term "availability" in the introduction and its importance in the context of water scarcity. The manuscript has been revised to provide a clearer explanation of what "availability" means in the context of usage rates, emphasizing its relevance to the paper's focus on water scarcity.

Comment 6: Finally, the abstract well describes the paper but it does not mention the key attributes of the WPI, the principle components or weighing factors, or the recommendations of the authors. I think this is mistake because that is the principal focus of the paper.

Response 6: The observation about the abstract not sufficiently highlighting key attributes of the WPI, principal components, weighing factors, and authors' recommendations has been considered. The abstract has been improved to better encapsulate these essential aspects, providing a more comprehensive summary of the paper's principal focus.

We are grateful for reviewer’s valuable feedback, which has significantly contributed to refining the manuscript. These improvements aim to enhance the overall clarity and accessibility of our research, and we look forward to the opportunity for further collaboration.

Best Regards,

Silvana Pacheco-Treviño and Mario Guadalupe Manzano-Camarillo

Corresponding Author: Silvana Pacheco-Treviño